# Casual Wage Labour, Food Security, and Sustainable Rural Livelihoods in Malawi

Hiroko Gono [1,*], Tsutomu Takane [1] and Dickson Mazibuko [2]

1   Faculty of International Agriculture and Food Studies, Tokyo University of Agriculture,
    Tokyo 156-8502, Japan
2   School of Natural and Applied Sciences, University of Malawi, Zomba P.O. Box 280, Malawi
*   Correspondence: hg203835@nodai.ac.jp; Tel.: +81-3-5477-2420

**Abstract:** Casual wage labour (known as *ganyu* in Malawi) is a widely adopted strategy to cope with insufficient income and food production in rural households. Although previous studies have discussed the magnitude of the contribution of *ganyu* to rural livelihoods, the actual conditions of individual rural households have not been studied in detail. The current research conducted a detailed village-level case study to analyse the relationship between *ganyu* and rural livelihoods in Malawi. The characteristics of three categories of households were examined: those that engaged in *ganyu*, those that employed *ganyu*, and those that engaged in and employed *ganyu*. The study found that: (1) income from *ganyu* and its contribution to household food security differed considerably based on age, gender, household circumstances, and local conditions; (2) households that employed *ganyu* were not necessarily wealthy or achieved self-sufficiency in maize production; and (3) contradictory behaviour of a household to engage in and employ *ganyu* was a result of the context-specific needs of that household. Rather than viewing *ganyu* as poorly paid agricultural wage labour, the study propose acknowledging that each household has its own rationale for engaging in or employing *ganyu*. Such an understanding from a household-level perspective would better inform poverty and food security policies.

**Keywords:** casual wage labour; food security; sustainable rural livelihood; food and cash shortages; *ganyu*; Malawi

## 1. Introduction

Although poverty and hunger problems in developing countries have improved since 1990, solving these problems remains an important issue for many countries in Sub-Saharan Africa (hereafter referred to as 'Africa') [1]. To achieve the Sustainable Development Goals and solve the problems of poverty and food insecurity, it is necessary to consider measures that focus on the people in such situations. The UN [2] states that although small-scale farmers are the backbone of agriculture, they are the most vulnerable group in rural areas and have significant differences in labour productivity and annual income compared to those of large-scale farmers. In addition, among small-scale producers, female-headed households are even more disadvantaged, with average annual incomes lower than those of male-headed households.

Currently, policies that target poverty and food insecurity among small-scale farmers in African countries focus mainly on improving agricultural productivity. A typical policy of such focus is the input subsidy program implemented after 2000 [3,4]. However, studies have shown that this policy does not target the most vulnerable households [5–8]; instead, politicians use it to gain votes [7,9,10], and its implementation under the conditions of low irrigation coverage does not contribute to increasing the agricultural productivity when compared to the financial burden [9]. In addition to the problems related to policy implementation, studies have found that the current agricultural policy is insufficient in terms

of enhancing sustainable agricultural productivity and adapting to climate change [11], and that income growth through increased agricultural productivity cannot be expected to continue in the long term [12]. Chinsinga et al. ([13], p. 5) argued that for many small-scale farmers, 'agriculture is no longer reliable as an exclusive means of subsistence and must be supplemented with other livelihood strategies'.

Taking all this information into account, this study analyses off-farm livelihood strategies adopted by small-scale farmers in rural Malawi through detailed village case studies. This study particularly focuses on casual wage labour called *ganyu* in Malawi. *Ganyu* is an opportunity to earn off-farm income, especially for rural households with no income source other than agricultural production activities. The study examines the role of *ganyu* in reducing food insecurity and poverty among rural households and clarifies the interrelationship between *ganyu* and other economic activities. With a focus on off-farm livelihood strategies, the study attempts to counterbalance the current policy orientation that mainly focuses on agricultural production in reducing poverty and food insecurity.

The argument of this study is based on a political economy analysis that incorporates a framework of sustainable rural livelihoods [14,15]. The political economy analysis, unlike the neoclassical economic analysis that adopts methodological individualism, emphasises that individual actions are conditioned by the entire economic systems. It tries to examine how wider structural forces that comprise social, economic, and political dimensions of human societies influence individual and household decision making. A critical issue for the political economy analysis is who has access to and control over productive resources such as land, labour and capital, and how the output obtained by productive activity is distributed among different social groups, households, and individuals. The distribution of productive resource and output is usually unequal, and the material structures of power shape livelihoods of people.

The framework of sustainable rural livelihood was first developed in the 1990s, and since then, this earlier version of framework was widely adopted by academics and policy makers [16–18]. Under the framework, sustainable livelihood is described as follows: 'A livelihood comprises the capabilities, assets (including both material and social resources), and activities required for a means of living. A livelihood is sustainable when it can cope with and recover from stresses and shocks, maintain or enhance its capabilities and assets while not undermining the natural resource base' ([16], p. 5). A notable feature of this framework is that it has a holistic analytical approach. The framework emphasises dynamic interrelations between different livelihood strategies adopted by small-scale farmers and often mediated by institutions and social relations.

While this earlier version of framework was adept at capturing diverse aspects of rural livelihoods, the approach's lack of adequate focus on the structural drivers of poverty, intra-household relations, historical forces and spatial dynamics of livelihood reproduction was a key weakness [15]. To overcome the weakness, a new framework that incorporated the political economy analysis was proposed [14,15]. The new framework attempts to capture the relationship between structures and processes (institutions, organizations, and policies) and rural livelihoods from an analytical perspective that captures the diversity of rural livelihoods—who, where, why, and how—as well as the differences in temporal and spatial dynamics. This new way of understanding of sustainable rural livelihood with political economy analysis allows us to move beyond the ambiguity of viewing changes as factors (an early version of rural livelihood framework) and simple cost–benefit incentives (narrow economistic frames) to capture the reality of rural households with a wider structural and dynamic perspective.

In Malawi, most of the poor population live in rural areas, and rainwater-dependent subsistence agriculture is their main livelihood. Most rural households are involved in small-scale farming, and the size of cultivated land per rural household is small (approximately 0.5 ha) [19]. The Malawian government has focused on improving the productivity of small-scale farms, but the food consumption and nutritional intake of rural households have not been improving [20]. The factor most responsible for this situation is the sea-

sonality of agricultural production, which is closely related to the food security of rural households and their vulnerability to poverty [21].

In rural African societies, the following behaviours can be observed under conditions of seasonal food insecurity: limiting consumption and expenditure (e.g., eating less, purchasing cheaper ingredients, and reducing non-food expenses) and obtaining additional food and cash by selling assets, engaging in casual labour, and borrowing [22]. These behaviours are observed in rural Malawi, where it is common for people to compensate for food and cash shortages by engaging in off-farm activities such as casual wage labour (*ganyu*). The compensation for *ganyu* can be paid in cash or in kind with the staple food maize; the latter form of compensation directly contributes to alleviating household food shortages.

Previous studies have highlighted the importance of off-farm activities in sustaining rural livelihoods in developing countries [23–26]. Among the off-farm activities, casual wage labour was found to be particularly important in African countries [27–29]. Previous studies on casual wage labour in Malawi have discussed the magnitude of the contribution of *ganyu* to rural livelihoods. Whiteside [30] indicated that the actual circumstances of *ganyu* vary by region and year, and the importance of *ganyu* differs according to the income level of rural households. Whiteside [30] also noted that measuring the significance of *ganyu* for rural residents using only quantitative data is difficult. *Ganyu* has also been shown to play an important role during food shortages for poor rural households [31,32]. However, a baseline survey conducted by an international non-governmental organisation reported that although >50% of the households facing food insecurity were engaged in *ganyu* to meet their food demands, this need was not satisfied by simply engaging in *ganyu* in many households [33].

Moreover, the role of *ganyu* in rural Malawi has implications that include the traditional social obligation of wealthy farmers to employ poor neighbouring households [34]. Therefore, rich farmers often hire more *ganyu* than they require [30]. Furthermore, households other than those with high incomes also employ *ganyu* [35], and Takane [36] presents a case where a household headed by an elderly female and with a scarcity of family labour was forced to hire a *ganyu* to supplement the labour shortage. In other words, the reality is that *ganyu* does not simply mean that wealthy farmers employ poor farmers [31]. Therefore, understanding *ganyu* in rural Malawi requires analysis of the characteristics captured by statistical variables and the actual conditions of individual rural households in detail.

Hence, this study clarifies the actual circumstances of *ganyu* in rural Malawi from a rural household-level perspective by focusing on the following points. The first is whether it is possible to procure sufficient maize, the staple food required by rural households, by engaging in *ganyu*. In Malawi, many farmers conduct subsistence farming, mainly of maize, in their fields. Approximately 46% of food and 60% of the necessary energy consumption are said to be derived from maize [37]. In other words, maize is the most important crop for Malawian farmers to support their livelihood and food consumption. *Ganyu* is a coping behaviour chosen by rural households under seasonal food shortages; therefore, examining whether it is possible to procure sufficient staple food by engaging in *ganyu* may provide a detailed understanding of the actual circumstances of farmers during times of food shortages.

The second is to accurately illustrate the concept of *ganyu* as perceived by Malawian farmers. Farmers perceive *ganyu* in various ways: the exchange of labour between relatives and neighbouring rural households, seasonal employment on large farms, and road construction and transportation work other than agricultural work. Hence, there are various methods of calculating the payment amount for labour, such as payments based on the amount of work completed, the content of the contracted work, and the number of days worked. However, many previous studies have limited the subject of *ganyu* to agricultural work supplied within a village or interpreted *ganyu* as piecework. In this study, the actual circumstances of *ganyu* in Malawi are revealed in more detail by analysing all the labour activities that farmers have responded to as being *ganyu*.

The third is to position *ganyu* in the context of individual rural households' diverse circumstances and local characteristics. Previous studies did not sufficiently investigate the characteristics of households that engage in or employ *ganyu* (e.g., age and gender composition, ownership of land and livestock assets, economic activities they engage in, and income) and whether engaging in *ganyu* can result in the procurement of sufficient food for rural households. Clarifying the relationship between *ganyu* and rural households also requires a detailed investigation regarding the circumstances under which households have engaged in or employed *ganyu*. In this study, we investigated the actual conditions underlying the use of *ganyu* and rural households in Malawi while showcasing individual cases of rural households and characteristics such as the existence or non-existence of economic opportunities unique to the region.

The significance of this study is twofold: one is contributing to the literature on sustainable rural livelihoods in developing countries. By providing a detailed examination of the interrelationship between the farmer's strategy of using casual wage labour and its effects on household food security, this study illustrates the complex and diverse situations of reality in rural livelihood. Another significance is that it informs policy making regarding poverty reduction and food security. A household-level perspective adopted in this study to understand the context-specific decision making of farmers provides a counterbalanced view to the dominant macro perspectives in policy making.

## 2. Materials and Methods

### 2.1. Overview of the Survey and Case Study Villages

This study adopted a mixed method that combined a household survey and case studies in selected villages. The survey was conducted to collect quantitative data such as income, land holding, and household demography, while the case study method was adopted to obtain qualitative data that were relevant for an in-depth understanding of context-specific livelihood strategies adopted by rural households [38,39].

The survey for this study was conducted in two villages in the northern region of Malawi (Villages C and Y) and one village in the southern region (Village E; Figure 1). Study villages were selected to reflect differences in environmental conditions, population pressure on land, social systems (patrilineal and matrilineal), economic activities, and access to information. These factors are influenced by the geographical conditions of the villages. Malawi's landmass stretches in the north–south direction from the low-altitude lakeside to high-altitude plateau areas. The population pressure on the land is overwhelmingly higher in the southern region than in the northern region [40]. Additionally, the social system consists of ethnic groups that are divided into the northern region, which adopts a patrilineal system, and the southern region, which adopts a matrilineal system. Given the differences between northern and southern Malawi, the current research decided to select study villages from both the northern and southern regions. In selecting the study villages in both regions, the researchers consulted with officers of the Ministry of Agriculture and selected villages in the north, where access to economic activity and information is good, and villages in the south, where access to economic activity and information is poor.

Chitipa District, to which the two northern villages (Villages C and Y) belong, is located 700 km from the capital city in Lilongwe District. Chitipa District borders Tanzania and Zambia. Villages C and Y are located near the area where the district government offices are located. In addition, Villages C and Y are 3.9 km and 3.0 km, respectively, from the town centre, which houses a permanent market of agricultural materials and foodstuffs, rendering relatively convenient access to information and economic activities. One village in the south (Village E) is located in Zomba District, which is 300 km from the capital in Lilongwe District and borders Mozambique. Unlike the two northern villages, Village E is located in a disadvantageous area, far from the district government offices area. In addition, permanent markets in the neighbourhood are lacking, and agricultural materials and foodstuffs can be purchased only at the market, which is open twice a week along the main road located at a distance of 6.0 km. As the village is 39.0 km from the district

government offices, government officials rarely visit the village, rendering inconvenience to residents in accessing information and economic activities.

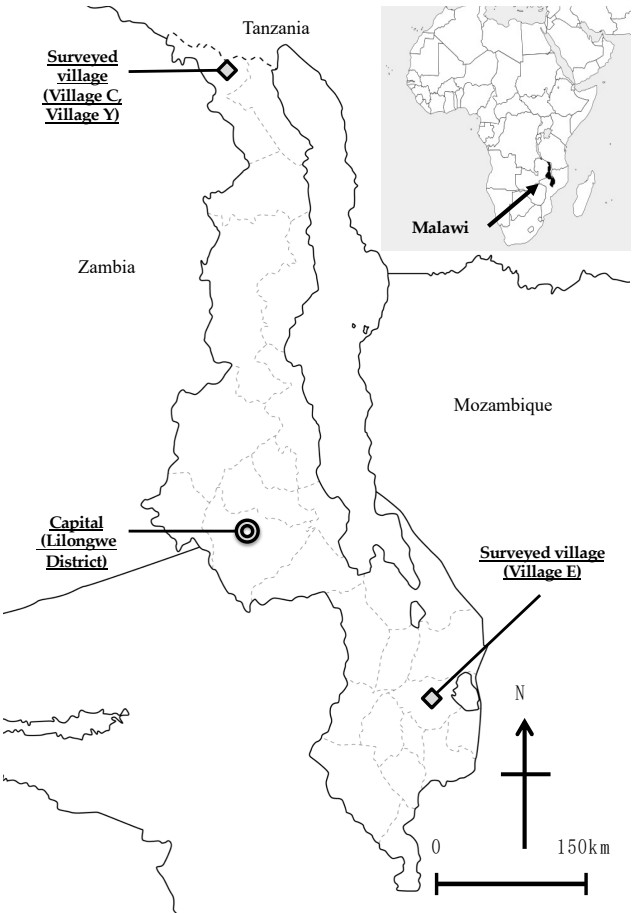

**Figure 1.** Location of surveyed villages. Source: Created by the authors.

Malawi has a rainy season once a year, and most farmers cultivate crops only during this period. Rain begins to fall in the southern region, and crops (mainly the staple food, maize) are cultivated in Village E from November to March and in Villages C and Y in the northern region from December to April. No machinery has been introduced for agricultural activities, and crop cultivation is instead centred on family labour.

This study conducted surveys in the two northern villages (Villages C and Y) in September 2015. The target households for the survey were selected via random sampling. The current study selected 21 households (of the 170 village households) in Village C and 20 households (of the 152 village households) in Village Y. The survey in the southern village (Village E) was conducted in August 2016, and 57 households (of the 85 village households) were interviewed. In Village E, a re-survey was conducted in August 2017 to target the same households interviewed in 2016.

Interviews with each household were based on a structured questionnaire and were conducted by survey assistants who were proficient in the local language and staff members of the Extension Planning Area under the jurisdiction of the Ministry of Agriculture. One of the authors was present to record all interviews. None of the households selected via random sampling refused to be interviewed, and therefore, nonresponse bias was avoided. In addition, to avoid late response bias, all interviews were conducted and completed within a few weeks during the off-farm season when farmers were not busy in farm work and readily available for interview. To avoid common method bias and accurately elicit household livelihood data, the authors attempted to avoid using difficult or ambiguous expressions in the questionnaire. In cases where responses contained uncertain quantities or

values (e.g., harvest volume and prices), data were crosschecked to guarantee the accuracy by consulting the coordinators (selected in each village) who were knowledgeable about villagers' activities or referring to previously researched local market price data. As part of the survey, Global Positioning System was used to measure the cultivated land area, which was determined by walking with the farmers around their cultivated land. In addition to the questionnaire, in-depth interviews were conducted to obtain qualitative data about why farmers decided to adopt particular livelihood strategies.

For this study, the surveyed households were classified into three categories: those that engaged in *ganyu*, those that employed *ganyu*, and those that both engaged in and employed *ganyu*. Table 1 shows the breakdown of each type in the villages that were surveyed.

**Table 1.** Number of households that engaged in *ganyu*, employed *ganyu*, and both engaged in and employed *ganyu*.

| Village C, 2013/2014 Number of Sampled Households 21 | | | Village Y, 2013/2014 Number of Sampled Households 20 | | | Village E, 2014/2015 Number of Sampled Households 57 | | | Village E, 2015/2016 Number of Sampled Households 57 | | |
|---|---|---|---|---|---|---|---|---|---|---|---|
| Engaged in *ganyu* | Employed *ganyu* | Both engaged in and employed *ganyu* | Engaged in *ganyu* | Employed *ganyu* | Both engaged in and employed *ganyu* | Engaged in *ganyu* | Employed *ganyu* | Both engaged in and employed *ganyu* | Engaged in *ganyu* | Employed *ganyu* | Both engaged in and employed *ganyu* |
| 13 | 8 | 4 | 10 | 7 | 2 | 50 | 6 | 3 | 53 | 6 | 5 |

Source: Created by the authors. Note: The households engaged in and employed *ganyu* was also counted among the households that engaged in *ganyu* and those that employed *ganyu*.

## 2.2. Method for Calculating the Secured Amount of Maize

The amount of maize procured by the surveyed households was calculated by subtracting the amount of maize sold from the sum of the amount produced in their own household, the amount purchased, and the amount paid in kind by *ganyu*. The amount of maize required for each household was calculated by considering the adult equivalent unit (AEU; 1 for men aged 15 years and older, 0.8 for women aged 15 years and older, and 0.5 for those aged below 15 years). Differences were considered in the amount of maize required because of dissimilarities in the household members' gender or age. Moreover, the amount of maize needed per AEU in 12 months was set at 200 kg (16.6 kg/AEU per month), based on the reports by Peters ([41], p. 18) and Gladwin ([42], pp. 181–182), and the amount of maize required for each household was determined based on this standard.

In rural Malawi, there are three ways to pay wages to *ganyu*: cash only, in kind (maize), and both cash and in kind. In addition to these payment methods, meals may also be provided. For calculating the amount of maize procured by households, the amount paid in kind that corresponded to the number of months' worth of maize required by each household was determined.

For cash payments, the study assumed that the entire amount was used to purchase maize; the amount of maize that could be purchased in the market was calculated, and the corresponding number of months' worth of maize was determined. When meals were provided, the monetary value in local currency (Malawian Kwacha: KW) of the provided meal was determined by asking about the details of the meals, the number of times they were provided, and how much the recipient would pay if they were to purchase those meals. After the monetary value of the provided meals was determined, the amount of maize that could be purchased in the local market with that amount was calculated, and the number of months' worth of maize this corresponds to was derived. Maize purchases were calculated based on local market price survey results for each surveyed year; since maize prices vary monthly, calculations were based on annual average prices.

## 3. Results and Discussion

### 3.1. Households That Engaged in Ganyu

First, the study investigated whether each household could procure the amount of maize equivalent to 12 months' worth of consumption by engaging in *ganyu*. Figure 2 shows the cumulative amount of maize produced within the household in terms of the number of months of consumption and the amount of maize that can be procured from the income earned by engaging in *ganyu*.

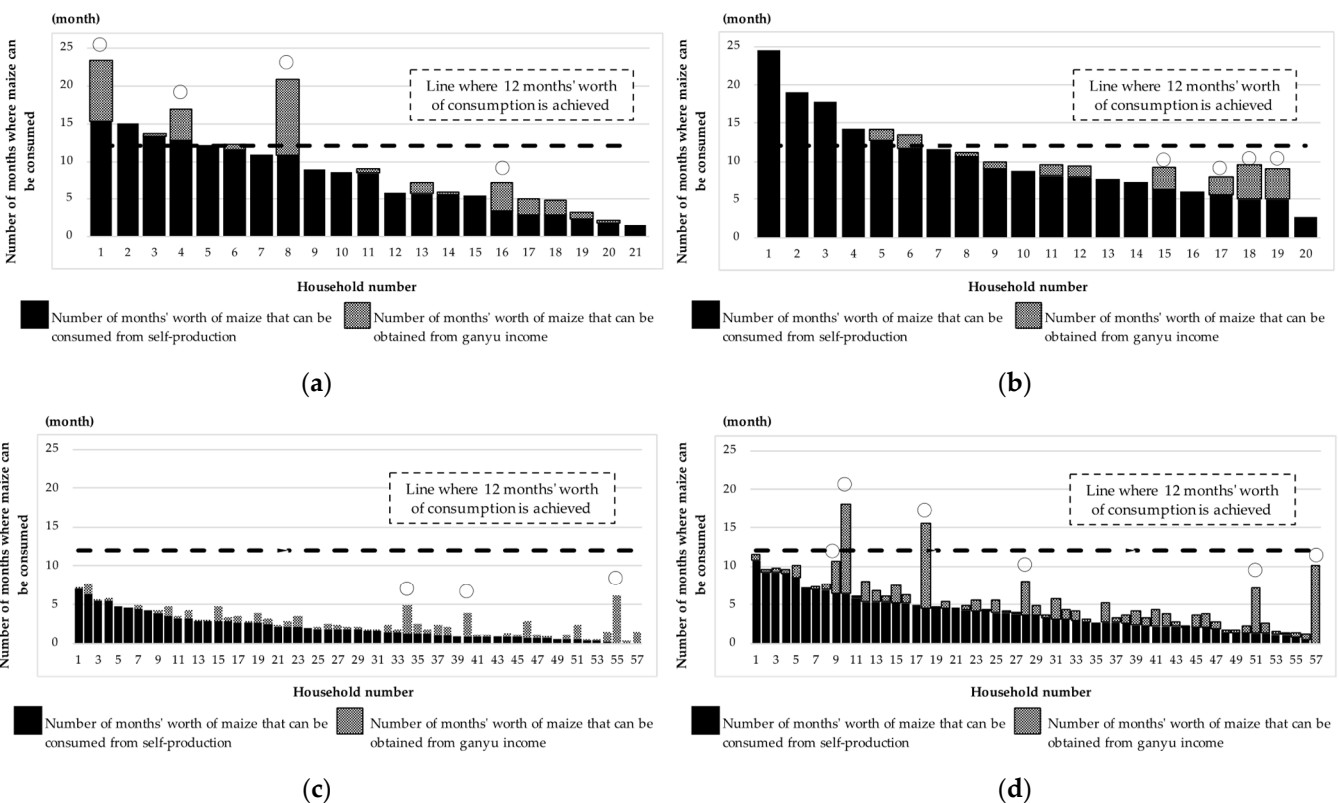

**Figure 2.** Amount of maize that could be secured from self-production and *ganyu* income: (**a**) Village C, 2013/2014; (**b**) Village Y, 2013/2014; (**c**) Village E, 2014/2015; and (**d**) Village E, 2015/2016. Source: Created by the author. Note: ○ indicates households that procured a relatively large amount of maize from their *ganyu* income.

According to Figure 2, most surveyed households cannot procure maize equivalent to 12 months' worth of consumption by simply engaging in *ganyu* to make up for the shortage of maize produced in their own households. The number of households that were able to make up for the shortage of maize by engaging in *ganyu* was as follows: two in Village C (representing 20% of households among those not achieving self-sufficiency), one in Village Y (representing 11% of households among those not achieving self-sufficiency), and two in Village E (representing 4% of households among those not achieving self-sufficiency (results from 2015/2016 season only)).

In the 2014/2015 season, in the case of Village E, none of the households managed to make up for the shortage in maize production by using the *ganyu* income to make purchases because of the erratic rainfall conditions and a sharp decrease in the amount of maize produced in that season. Moreover, the average market price of maize in the 2014/2015 season increased to 1.4 times that of the previous year because of the low maize production. Furthermore, there was no change in the wages received by households engaged in *ganyu* (No significant difference was found in average *ganyu* income between the 2014/2015 and the 2015/2016. In other words, the obstacle to the purchase of maize, which was in short supply due to poor production, was not the decline in *ganyu* wages

but rather the increasing market price of maize. Whiteside [30] indicated that in years of poor weather, the number of households seeking *ganyu* work increases, thereby decreasing the remuneration paid for *ganyu* (in cash or in-kind); however, we did not observe this phenomenon in this study.

Nevertheless, households with a relatively large amount of maize that could be procured with *ganyu* income were found in all the surveyed villages (Figure 2, circled households). Therefore, in this section, the study has discussed the characteristics of households that could procure a large amount of maize with their *ganyu* income for each surveyed village.

First, the study has considered Village C, with four households that could procure relatively large amounts of maize with their *ganyu* income (Figure 2a, household numbers 1, 4, 8, and 16). Among these, three households (household numbers 1, 4, and 16) were engaged in *ganyu* work other than agricultural work, which involved the construction of houses and roads. The increase in the income of these households is attributed to the large amount of work they undertook.

Table 2 shows the job content, average income, and the number of working days in which rural households were engaged. In Village C, the number of working days spent building houses and roads was more than that on other jobs. The average income per person per day obtained from carrying water needed for building blocks was lower than that obtained from construction work, although a similar number of working days was spent on both jobs in the village. The amount of *ganyu* payment is determined at the employer's discretion according to the amount of work requested; thus, we concluded that construction requires more labour than the work of carrying water for making blocks.

Furthermore, the amount of work requested differed by gender and age. In Village C, household number 8 could procure a large amount of maize using *ganyu* income. In this household, several members were engaged in *ganyu*, and although all of them were engaged in preparing the land for maize cultivation, their payments differed. In the case where an elderly female household head and the grandsons who attended junior and senior high school worked, the income of the latter from *ganyu* was about twice the income of the former, although they both worked the same number of days (six days).

Next, the study considered Village Y. The *ganyu* observed for house construction in Village C was also observed in one case in Village Y. The household that engaged in house construction (Figure 2b, household number 19) also received a large amount of maize from their *ganyu* income compared to that received by others within the village. However, in Village Y, compensation was not paid for the amount of work but for the skills of the workers. Although the average income of this household was high (5333 KW/person/day), the number of working days was smaller than the average of three days in Village C. Moreover, this was house construction work. It did not involve stacking blocks but roof building and painting. The male head of the household was engaged in this work. Although he did carpentry when requested by his boss, he did not have any such work in the year when the survey was conducted. Hence, the current study believes that he was entrusted with work that differed from that of other workers and for which he was paid a large amount.

Three households in Village Y (Figure 2b, household numbers 15, 17, and 18) procured a relatively large amount of maize using their *ganyu* income. These households were engaged in building blocks (Figure 2b, household numbers 15 and 17) and carrying water for building blocks (Figure 2b, household number 18). Village Y tended to have a higher average income from building blocks. Furthermore, the average daily income per person from carrying water for building blocks was high. They earned a large amount of *ganyu* income because they worked for 48 days.

Finally, the study considered the case of Village E. The households in Village E tended to engage in *ganyu* that were unique to the area where the village was located and obtained large amounts of maize. In Village E, nine households obtained relatively large amounts of maize from *ganyu* income in the 2014/2015 and 2015/2016 seasons (Figure 2c, household

numbers 34, 40, and 44; Figure 2d, household numbers 9, 10, 18, 28, 51, and 57), of which eight were engaged in *ganyu* that was unique to Village E, as described below.

The first *ganyu* unique to Village E was fishing using the dragnet. Of the households with relatively large amounts of maize that could be procured with *ganyu* income, three (Figure 2c, household number 55; Figure 2d, household numbers 18 and 57) went to Lake Chilwa, located approximately 40 km from the village, and engaged in *ganyu* involving dragnet fishing. As this *ganyu* does not depend on seasons, it can be conducted throughout the year. Therefore, these households went to Lake Chilwa for dragnet fishing *ganyu* almost every month throughout the year. Households that engaged in dragnet fishing *ganyu* stayed and worked for 3–7 days per engagement, indicating that households with more labour engagements had longer working days per year. Moreover, the average income per person per day was high, indicating that large amounts of maize could be procured from the income of households with several labour engagements.

The second *ganyu* unique to Village E was the land preparation *ganyu* in Mozambique, near the Malawi border. Five households (Figure 2c, household number 40; Figure 2d, household numbers 9, 10, 28, and 51) were involved in this *ganyu*, which is common in this region. This is because the population pressure on the land differs between Malawi and Mozambique. Malawi, which has high population pressure, has land and food shortages. In contrast, the Mozambique side has low population pressure on land and has labour shortages and markets for selling crops [30]. According to interviews conducted in the village, a large amount of cultivated land is available on the Mozambique side, where *ganyu* was easier to find, and the market price of maize was lower than that in the village. Village E is approximately 60 km from the Mozambique border, and the households involved in *ganyu* in Mozambique stated that it took them 1–3 days to travel by bicycle. The reason underlying the difference in the travel period to the place of *ganyu* employment was that although individuals could reach the border in one day, they sometimes went to villages further away from the border if *ganyu* was not found close to the border (Whiteside [43] reported that Malawian farmers travelled to rural villages in Mozambique 100 km from the border for *ganyu*). Many households spend more days on *ganyu* (agricultural work) in Mozambique than they do in their village, and as the income earned per person per day is high, they can make more income.

The number of households that could engage in high-income *ganyu* was limited. As shown in Table 2, cases of *ganyu* related to road and house construction work, as observed in Villages C and Y, were not observed in Village E. In addition, the number of cases of involvement in construction in each northern village was small, and it was apparent that all households could not be engaged in construction *ganyu*. The same was true for the dragnet fishing *ganyu* in Village E. The dragnet fishing *ganyu* is unique to areas with lakes, and there were only a few cases of such engagement, even in Village E. In other words, although these *ganyu* were high-income jobs, they were not available to all households who wanted to engage in this work. Moreover, for large *ganyu* income, the households had to work for long periods. The number of working days for construction, carrying water for building blocks, and dragnet fishing was greater than that for the other types of work. This indicated that household members that could be away from home for long periods (i.e., men) tended to work more than others. In fact, in the surveyed villages, all the workers who engaged in these tasks were men, except for carrying water to building blocks (Only the case of Village Y case involved a female. The reason this household was able to engage in *ganyu* for long periods was because of the ages of the dependent children (19, 17, 13, and 7 years)). In other words, to procure large amounts of maize with *ganyu* income, one must engage in long-term *ganyu* work that often involves long-distance travelling.

In summary, households that could procure large amounts of maize with *ganyu* income were those that were engaged in *ganyu* that was not agricultural work but instead in *ganyu* that required long periods of work and travelling. Among households engaged in jobs other than agricultural work, those engaged in construction work in the two northern villages and fishing using the dragnet in the southern village had high average incomes. Moreover,

although some households earned a large income from land preparation related to maize cultivation, the income differed considerably based on age and gender. Depending on the job opportunities in the area, gender and age of the household members, and household circumstances, some households could participate in *ganyu*, and others could not. Although these results could be understood to favour healthy or young men, they should not be interpreted as favouring men since women can engage in long-term *ganyu* too, depending on the household members.

### 3.2. Households That Employed Ganyu

Next, the study investigated the amount of maize procured by households that employed *ganyu*. Figure 3 shows the number of months' worth of maize produced in each household that corresponded to the payment to *ganyu*. Maize that was paid in kind was shown in terms of the number of months' worth of maize produced in the household (upper graph), and cash payments and meals provided are shown in terms of the number of months' worth of maize if purchased at a market (lower graph). Regarding cash payments and meals provided, the study calculated the percentage of each household's income (excluding the amount consumed by the household) and showed the proportion on the graph.

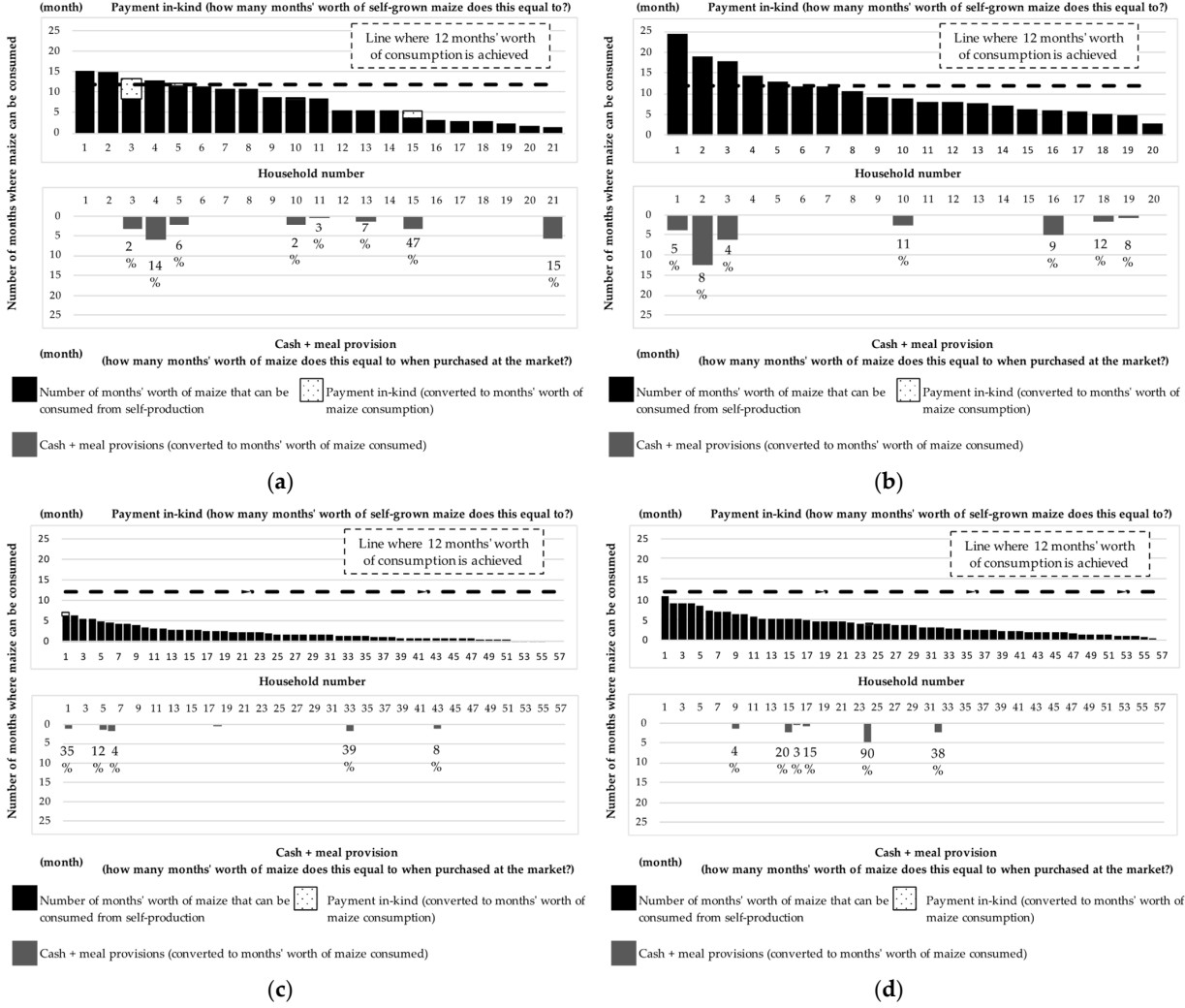

**Figure 3.** Amount of self-grown maize equivalent to *ganyu* payment: (**a**) Village C, 2013/2014; (**b**) Village Y, 2013/2014; (**c**) Village E, 2014/2015; and (**d**) Village E, 2015/2016. Source: Created by the author. Note: The proportions shown in the figure indicate the *ganyu* payment (cash + meals provided) from the household income (excluding maize consumed in that household).

According to the 2015/2016 season results in Villages C and E, one household in each village contributed over half of their household income to *ganyu* payment despite not achieving self-sufficiency in maize production (Figure 3a, household number 15; Figure 3d, household number 24). These households would be closer to meeting their 12-month requirement of maize if they used the payment allocated for *ganyu* for consumption in their own households. The household in Village C (Figure 3a, household number 15) would be able to procure 8.6 months' worth of maize if they did not hire *ganyu* and used the money for self-consumption. Similarly, the household in Village E in the 2015/2016 season (Figure 3d, household number 24) would be able to procure 9.1 months' worth of maize.

These two households had common characteristics: the household head was elderly, and the amount of livestock holding assets was large. Table 3 shows the characteristics of households that employed *ganyu* in the 2015/2016 season in Villages C and E. First, the livestock holding assets of the household in Village C were worth 31,800 KW (862 USD at the 2013 rate), making it the household with the largest livestock holding asset in the village. This household's livestock-holding assets were by far the largest in the village, which was approximately 3.5 times that of the household with the second-largest livestock-holding asset in the village. The household head was 77 years old, but the household labour force was larger than the average in the village because more older members were living together than in the other households. However, owing to the large area of maize-cultivated land, the household labour force per cultivated area was 1.6 people/ha, which was lower than the village average. Similar to that observed in Village C, the livestock holding assets of the household in Village E were the largest in the village (105,720 KW = 287 USD at the 2013 rate), which was approximately 2.6 times that of the household with the second-largest livestock holding assets in the village. Although the household labour force and labour force per cultivated land area were not smaller than those of the households employing *ganyu* in the same village, the household labour force per cultivated area was less than the village average.

The results indicated that these two households employed *ganyu* to supplement labour shortages, as mentioned in a previous study [36], and highlighted the traditional social obligations for wealthy farmers to employ poorer neighbouring households [34]. Both households had several large livestock, such as cows and pigs, raised in livestock sheds, which was rare in the villages. Therefore, we believe many people visited these houses in search of a job as *ganyu* because the size of the livestock assets could be easily deemed from the house exterior. However, although the livestock assets of these households were large, their income was low. Therefore, the proportion of payments towards *ganyu* from the household income was large. Although these households could supplement the labour force needed for maize production by employing *ganyu*, they were forced to employ *ganyu* based on social obligations while not achieving maize self-sufficiency, thereby increasing the possibility of reducing the amount of maize consumption or livestock-holding assets.

**Table 2.** Details of the *ganyu* work engaged in by rural households.

| *Ganyu* Work Content | Village C, 2013/2014 Number of Cases (Total) 20 | | | Village Y, 2013/2014 Number of Cases (Total) 14 | | | Village E, 2014/2015 Number of Cases (Total) 84 | | | Village E, 2015/2016 Number of Cases (Total) 78 | | |
|---|---|---|---|---|---|---|---|---|---|---|---|---|
| **Agricultural** | Number of cases (Proportion) | Average income (KW/person/day) | Average number of working days (Days) | Number of cases (Proportion) | Average income (KW/person/day) | Average number of working days (Days) | Number of cases (Proportion) | Average income (KW/person/day) | Average number of working days (Days) | Number of cases (Proportion) | Average income (KW/person/day) | Average number of working days (Days) |
| Land preparation (maize) | 45% | 792 | 7 | 50% | 581 | 13 | 60% | 586 | 10 | 63% | 759 | 13 |
| <In village> | 45% | 792 | 7 | 50% | 581 | 13 | 46% | 409 | 8 | 36% | 487 | 8 |
| <In Mozambique> | 0% | NA | NA | 0% | NA | NA | 13% | 1216 | 18 | 27% | 1122 | 19 |
| Weeding (maize) | 20% | 615 | 8 | 7% | 583 | 6 | 24% | 434 | 2 | 22% | 404 | 4 |
| Ridging (maize) | 0% | NA | NA | 0% | NA | NA | 5% | 618 | 2 | 3% | 263 | 8 |
| Harvesting (maize) | 5% | 1000 | 1 | 0% | NA | NA | 0% | NA | NA | 0% | NA | NA |
| Cleaning up residue (maize) | 0% | NA | NA | 0% | NA | NA | 0% | NA | NA | 0% | NA | NA |
| Work on crops other than maize | 0% | NA | NA | 0% | NA | NA | 0% | NA | NA | 0% | NA | NA |
| Total | 70% | NA | NA | 57% | NA | NA | 88% | NA | NA | 87% | NA | NA |
| **Non-agricultural** | Number of cases (Proportion) | Average income (KW/person/day) | Average number of working days (Days) | Number of cases (Proportion) | Average income (KW/person/day) | Average number of working days (Days) | Number of cases (Proportion) | Average income (KW/person/day) | Average number of working days (Days) | Number of cases (Proportion) | Average income (KW/person/day) | Average number of working days (Days) |
| Building blocks | 5% | 250 | 10 | 29% | 1849 | 17 | 1% | 834 | 30 | 4% | 827 | 11 |
| Carrying water (for building blocks) | 5% | 505 | 24 | 7% | 521 | 48 | 5% | 320 | 11 | 3% | 504 | 9 |
| House construction work | 15% | 1167 | 25 | 7% | 5333 | 3 | 0% | NA | NA | 0% | NA | NA |
| Road construction work | 5% | 961 | 21 | 0% | NA | NA | 0% | NA | NA | 0% | NA | NA |
| Fishing using dragnet | 0% | NA | NA | 0% | NA | NA | 5% | 942 | 40 | 5% | 941 | 58 |
| Making toilet (digging holes) | 0% | NA | NA | 0% | NA | NA | 1% | 832 | 2 | 0% | NA | NA |
| Harvesting fertile soil | 0% | NA | NA | 0% | NA | NA | 0% | NA | NA | 1% | 202 | 5 |
| Total | 30% | NA | NA | 6 (43%) | NA | NA | 12% | NA | NA | 13% | NA | NA |

Source: Created by the authors. Note: All average incomes are converted using the rural consumer price index (RCPI), with that of 2013/2014 set as 100. In 2013, 1 USD = 369 KW (Malawian Kwacha). The minimum wage per day for the rural region in July–December 2013, set by the government of Malawi, was 317 KW [44].

**Table 3.** Characteristics of households that employed *ganyu*.

**Village C, 2013/2014**

| Household number (Figure 3a) | Proportion of *ganyu* payments out of household income | Gender of the head of household | Age of head of the household | | Number of household members | | Labour force | | Area of maize-cultivated land | | Household labour force per area of maize-cultivated land | | Livestock holding assets per household | | Total income per AEU | |
|---|---|---|---|---|---|---|---|---|---|---|---|---|---|---|---|---|
| | | | (Years) | Village ranking | (People) | Village ranking | (People) | Village ranking | (Ha) | Village ranking | (People/ha) | Village ranking | (KW) | Village ranking | (KW/AEU) | Village ranking |
| 3 | 2% | F | 57 | mid | 3 | low | 1.0 | low | 0.6 | mid | 1.7 | low | 0 | low | 181,463 | high |
| 4 | 14% | M | 35 | low | 5 | mid | 2.0 | low | 0.8 | high | 2.6 | low | 0 | low | 57,245 | high |
| 5 | 6% | F | 68 | high | 2 | low | 1.0 | low | 0.3 | low | 3.4 | low | 0 | low | 54,453 | high |
| 10 | 2% | M | 40 | low | 4 | low | 2.5 | mid | 0.6 | mid | 4.0 | low | 6500 | low | 175,414 | high |
| 11 | 3% | M | 40 | low | 7 | high | 3.0 | mid | 0.6 | mid | 5.2 | mid | 13,150 | mid | 15,318 | low |
| 13 | 7% | M | 59 | mid | 7 | high | 5.0 | high | 0.6 | mid | 8.3 | high | 17,600 | mid | 22,311 | mid |
| 15 | 47% | M | 77 | high | 7 | high | 4.0 | high | 2.5 | high | 1.6 | low | 318,000 | high | −19,834 | low |
| 21 | 15% | F | 91 | high | 3 | low | 1.5 | low | 0.3 | low | 4.5 | mid | 1000 | low | 60,935 | high |
| Village average | | | 57 | 1 > 21 | 6 | 1 > 21 | 3.2 | 1 > 19 | 0.6 | 1 > 20 | 7.5 | 1 > 21 | 27,164 | 1 > 14 | 44,414 | 1 > 21 |

**Village E, 2015/2016**

| Household number (Figure 3d) | Proportion of *ganyu* payments out of household income | Gender of the head of household | Age of head of the household | | Number of household members | | Labour force | | Area of maize-cultivated land | | Household labour force per area of maize-cultivated land | | Livestock holding assets per household | | Total income per AEU | |
|---|---|---|---|---|---|---|---|---|---|---|---|---|---|---|---|---|
| | | | (Years) | Village ranking | (People) | Village ranking | (People) | Village ranking | (Ha) | Village ranking | (People/ha) | Village ranking | (KW) | Village ranking | (KW/AEU) | Village ranking |
| 9 | 4% | M | 32 | low | 4 | mid | 2 | mid | 0.3 | low | 7.2 | high | 5375 | mid | 68,236 | high |
| 15 | 20% | M | 53 | high | 3 | low | 2 | mid | 0.6 | mid | 3.3 | mid | 2822 | low | 14,174 | |
| 16 | 3% | M | 33 | mid | 6 | high | 2 | mid | 0.4 | mid | 4.9 | mid | 10,078 | high | 35,591 | high |
| 17 | 15% | M | 75 | high | 7 | high | 1 | low | 2.3 | high | 0.4 | low | 5375 | mid | 8696 | low |
| 24 | 90% | M | 70 | high | 5 | mid | 2.5 | high | 0.7 | high | 3.7 | mid | 105,720 | high | 3584 | low |
| 32 | 38% | M | 56 | high | 5 | mid | 3.5 | high | 0.5 | mid | 6.4 | high | 0 | low | 8544 | low |
| Village average | | | 44 | 1 > 57 | 5 | 1 > 57 | 2.3 | 1 > 57 | 0.7 | 1 > 56 | 5.0 | 1 > 57 | 6633 | 1 > 33 | 23,794 | 1 > 57 |

Source: Created by the authors. Note: (1) For the proportion of *ganyu* payments from the household income, the income from self-produced maize does not include self-consumed maize. (2) *Ganyu* payments represent amounts paid in cash and meals. (3) Labour force is denoted as follows: household members aged 16–64 years = 1, household members aged 13–15 years and 65 years or older = 0.5, and household members aged 12 years or younger = 0. (4) Adult equivalent unit (AEU is denoted as follows: male individual aged 15 years or older = 1, female individual aged 15 years or older = 0.8, and individual aged less than 15 years = 0.5). (5) The livestock holding assets per household and total income per AEU are converted using the rural consumer price index (RCPI), with 2013/2014 values set as 100. In 2013, 1 USD = 369 KW (KW means Malawian Kwacha). (6) The ranking levels in the village are shown in three levels from highest to lowest: high, mid, and low. (7) Households highlighted in grey are mentioned in the text.

*3.3. Households That Both Engaged in and Employed Ganyu*

Finally, the study considered the households that both engaged in and employed *ganyu*. In terms of labour supply and demand, this type of household both supplied labour for other households and demanded labour for its own farming. In terms of income, while the households earned cash and maize by engaging in *ganyu*, they also spent cash and maize by employing *ganyu*.

To investigate the cause behind such apparently contradictory behaviours, all 14 households that were both engaged in and employed *ganyu* were classified into the following four groups according to the household labour force or household labour force per cultivated land area and the size of household income; then, the study investigated the household circumstances in detail (Table 4 and examples are described later):

1. Households with low or household labour force per cultivated land area and high-income level.
2. Households with low or household labour force per cultivated land area and low-income level.
3. Households with sufficient labour or household labour force per cultivated land area and high-income level.
4. Households with sufficient labour or household labour force per cultivated land area and low-income level.

3.3.1. Households with Low Labour Force or Household Labour Force per Cultivated Land Area and High Level of Income

A common characteristic of the two households in this category (details mentioned below) is that the household head is the sole member of the labour force; therefore, they hired *ganyu* to supplement the labour force. The reasons for the labour shortage were not only the small labour force in the household (Village C: household number 3) but also the household members, including young children and wives, who could not work (Village C: household number 4). Moreover, the household head, who represented the main labour force, was engaged full-time in non-agricultural work; as such, it was difficult for them to engage in agricultural work in distant cultivated lands.

Moreover, the two households in this category achieved maize self-sufficiency and had high incomes; hence, the reasons for engaging in *ganyu* to deal with food shortages did not apply. However, in the second case below, the purpose was to obtain the agricultural inputs necessary for maize production. In other words, there were cases where households engaged in *ganyu* for investment to improve productivity, not for compensation for food shortages.

<Village C: Details of household number 3 (Table 4)>

This household (Village C, household number 3) had only one household member as part of the labour force and was the household with the least labour force (ranking 19/19) and highest income (ranking 1/21) in the village. This household consisted of three members: a woman (57 years old), the household head, and two grandchildren (aged 10 and 4). The main labour force comprised only the household head, who worked as a caretaker for female students at a private secondary school every day (365 days) and sold self-made doughnuts on the school grounds (288 days), which was responsible for the high income of the household. The farmland owned by this household, which was close to the house, was very small (0.01 ha); however, the household also owned 0.6 ha of land, gifted by the household head's parents, in a neighbouring village 4 km away, which was the centre of maize production for this household. This household employed *ganyu* in the distant land for land preparation and weeding.

<Village C: Details of household number 4 (Table 4)>

This household (Village C, household number 4) had two members as part of the labour force, which was a relatively small labour force within the village (ranking 16/19) and the fifth highest income in the village (ranking 5/21). As this household engaged in

*ganyu* other than agricultural work (road construction) for long periods at a high labour cost per day (as previously mentioned in the category of households engaged in *ganyu*), it earned a large amount of income from *ganyu*. Moreover, the *ganyu* payment for this household was in the form of in-kind payment of agricultural inputs (chemical fertilisers and maize seeds). This household consisted of five members: the household head (35 years old), his wife (32 years old), and their three children (aged 8, 5, and 2 years). The main labour force in this household was represented by the head, who worked as a *ganyu* for 21 days and as a night guard 327 days a year (four days off a month). This household owned a small amount of cultivated land measuring 0.1 ha near the house and rented 0.7 ha of land in a neighbouring village 4 km away for maize cultivation. They employed *ganyu* for land preparation, weeding, and harvest transportation work on the land. The household head and his wife comprised the labour force; however, as this household had two children aged five years or younger, it was difficult for the wife to go frequently to the farmland 4 km away for farm work.

### 3.3.2. Households with Low Labour Force or Household Labour Force per Cultivated Land Area and Low Level of Income

One household in this category (Village Y: household number 19) had a large family labour force; however, there was a shortage of household labour force per area of cultivated land owing to the large size of the cultivated land. Therefore, they used part of the income earned from engaging in *ganyu* (construction work) to employ *ganyu* to be used in distant fields.

Another household (Village E 2014/2015 season: household number 18) had labour shortage because the household head was elderly and female. Moreover, the household members who employed *ganyu* differed from those who were engaged in *ganyu*. In such cases, it is unclear whether the income from the *ganyu* was used to support the livelihoods of the household as a whole or was used for personal purposes.

<Village Y: Details of household number 19 (Table 4)>

This household's labour force comprised 6.5 household members, which was relatively high in the village (ranking 4/19). However, because of the large area of maize-cultivated land, the household labour force per cultivated land area was the lowest in the village, at 2.7 people/ha (ranking 19/19). As discussed in a previous subsection, this household had *ganyu* for whom employers paid a large amount for their skills, and the income earned from *ganyu* was high (ranking 2/11), although the total income was low within the village (ranking 19/20). This household consisted of nine people: the male household head (49 years old), his wife (47 years old), and seven children (24, 22, 19, 16, 13, 10, and 7 years old). It owned 0.4 ha of cultivated land near the house and 2.0 ha of cultivated land 2.0 km away from the house (within the same village), making it one of the largest households in the village (ranking 2/20). Although the household head was a carpenter (as discussed in a previous subsection, Section 3.1), he was engaged in construction work *ganyu* in the year when the interview was conducted because he had no carpentry-related requests from his boss during that time, and he earned a large amount of *ganyu* income. This household employed a *ganyu* to prepare the cultivated land (2.0 ha) away from the house. More than half of this household's income is *ganyu* income, suggesting that the amount of payment for hiring *ganyu* was covered by the large amount of income that the household head earned from engaging in *ganyu*.

<Village E 2014/2015 season: Details of household number 18 (Table 4)>

This household's labour force comprised 1.5 members, which was relatively low in the village (ranking 49/57); the income was also relatively low (ranking 53/57). This household had an elderly (68 years old) female household head with two grandchildren (aged 17 and 12 years) living with her to help out. Agriculture was the household's sole income source; hence, the income was also relatively low. The 17-year-old grandchild (male) engaged in *ganyu*, particularly field land preparation work in a neighbouring household. Furthermore,

the household employed *ganyu* to prepare the 0.7 ha cultivated land near the house. In this household, considering that it was the young grandchildren and not the household head who engaged in *ganyu*, the income earned by engaging in *ganyu* was small (ranking 45/51). Engaging in *ganyu* was possibly to support livelihoods, although making pocket money for the grandchildren could be another plausible purpose.

### 3.3.3. Households with Sufficient Labour Force or Household Labour Force per Cultivated Land Area and High Level of Income

Two households with this characteristic (Village E 2015/2016 season: household numbers 9 and 16) had children or toddlers among the household members, and the wife might not be able to work. Moreover, a characteristic of these two households was that the person employed as a *ganyu* was a relative. In Malawi, people sometimes ask wealthy relatives for help in times of food or cash shortages. Prioritisation of relatives over other villagers as *ganyu* may indicate labour employment for social assistance.

<Village E 2015/2016 season: Details of household number 9 (Table 4)>

This household's labour force comprised 2.0 people, a relatively moderate position within the village (ranking 21/57), and the income was the fifth highest in the village (ranking 5/57). As discussed in a previous subsection (Section 3.1), this household was engaged in the unique *ganyu* observed in Village E (preparation of land in Mozambique), and it could procure large amounts of maize from *ganyu* income. This household consisted of a male household head (32 years old), his wife (33 years), and two children (6 and 4 years). The household head was the main labour force of this household, and other than working as a *ganyu* for 18 days, he ran a self-employed business, selling cheaply purchased beans at the market from June to October (120 days). The cultivated land of this household included not only the 0.1 ha of cultivated land owned by the household but also 0.2 ha of cultivated land rented in the neighbouring village 2 km away. This household employed *ganyu* for ploughing, levelling and weeding on the leased farmland. Villagers living nearby were employed for land preparation work, and relatives were hired for weeding. This household's labour force comprised the household head and his wife. However, the two dependent children were young, indicating that it was difficult for the wife to engage frequently in farm work on the leased farmland 2 km away. Meanwhile, the busiest period for the head of this household was until October, which was before the rainy season. Therefore, it was possible for them to engage in agricultural work in their own field from November to January, when weeding work was in demand. In other words, the weeding work performed by the relatives could also have to be performed by the household head himself.

<Village E 2015/2016 season: Details of household number 16 (Table 4)>

This household's labour force comprised 2.0 people, which was of moderate level within the village (ranking 21/57), and the income was the 11th highest in the village (ranking 11/57). This household consisted of the male household head (33 years old), his wife (29 years old), and four children (11, 8, 6, and 1 year(s) old). The head of this household was self-employed and operated a year-round bicycle repair business. Both the household head and his wife were engaged in *ganyu* in the same household in the same village; the household head was engaged in land preparation work for seven days, while the wife was engaged in weeding for three days. This household owned 0.2 ha of cultivated land near the house and 0.2 ha of cultivated land in a neighbouring village 3 km away; they hired relatives as *ganyu* to do the land preparation work on the cultivated land they owned in the neighbouring village. It was challenging to go to the farmland 3 km away and frequently do farm work while caring for the six- and one-year-old children.

### 3.3.4. Households with Sufficient Labour Force or Household Labour Force per Cultivated Land Area and Low Level of Income

Households in this category would normally only need to engage in *ganyu* to cover up for cash and maize shortages. However, there was a special circumstance in which a household in this category (Village Y: household number 18, Village E 2015/2016 season: household number 32) had school-going children. The school attendance of working-age children caused a shortage in the household labour force. However, the female head of this household could engage in *ganyu* for a long period because of the age of the dependent children. In another household with a male household head, it was the sons who were engaged in *ganyu*, and the household members who employed *ganyu* differed from those who were engaged in *ganyu* (same case of Village E 2014/2015 season: Details of household number 18).

<Village Y: Details of household number 18 (Table 4)>

This household's labour force comprised 3.5 people, and although it had a relatively small labour force in the village (ranking 14/19), its labour force per cultivated land area at 5.8 people/ha (ranking 7/19) was at an intermediate to a high level. Moreover, its income was the lowest in the village (ranking 20/20). However, as discussed in a previous subsection (Section 3.1), this household was the only one able to obtain large amounts of maize from *ganyu* (carrying water for building blocks) despite having a female household head. One characteristic of this household was that the ages of the dependent children who were part of the household were high (19, 17, 13, and 7 years old). As the dependent children in this household were relatively older, the household head could leave their home and engage in *ganyu* for long periods (48 days) despite being a woman (the number of days engaged in *ganyu* was the largest in the village). Another characteristic of this household was that the house they lived in was not in the village but in the centre of the town (3.0 km away). However, they used the cultivated land owned in Village Y. This household employed *ganyu* to weed the land in Village Y. Its labour force comprised three school-going children (19, 17, and 13 years old) in addition to the female household head, who represented the main labour force. As the children were school-going, they did not constitute a major part of the labour force; thus, the household had no choice but to employ labour that was in short supply in the household.

<Village E 2015/2016 season: Details of household number 32 (Table 4)>

This household had a relatively high labour force at 3.5 people (ranking 7/57) and a high labour force per cultivated land area at 6.4 people/ha (ranking 16/57), while income was at a low level in the village (ranking 45/57). This household consisted of a male household head (56 years old), his wife (59 years old), and three school-going children (18, 14, and 12 years old). The labour force in this household was large because two children were included in the labour force; however, the main labour force in this household was the head of household and his wife. The household cultivated maize and vegetables on different plots of land, with maize cultivated around the house and vegetables cultivated in a neighbouring village 3 km away. In this household, vegetable cultivation was mainly carried out by the male head, suggesting that *ganyu* was hired to compensate for the lack of labour in the household that was required for maize cultivation. In addition, the son from this household was engaged in *ganyu* in the form of land preparation during his secondary school vacation. This is the same situation as that of 'Village E 2014/2015 season: Detail of household number 18', as engaging in *ganyu* was possibly to support livelihoods, although making pocket money for the son could be another plausible purpose.

**Table 4.** Characteristics of households that both engaged in and employed *ganyu*.

| | (a) Household Number | (b) Usage of *ganyu* for Far Away Cultivated Land | (c) Gender of the Head of Household | (d) Age of the Head of Household | (e) Number of Household Members | (f) Labour Force | (g) Area of Maize-Cultivated Land | (h) Household Labour Force per Area of Maize-Cultivated Land | (i) Livestock Holding Assets per Household | (j) Number of Days Engaged in Non-Agricultural Work | (k) Number of Days Engaged in *ganyu* | (l) Total Income per AEU | (m) Income from Non-Agricultural Work per AEU | (n) *Ganyu* Income per AEU | Villager | Relative | Both |
|---|---|---|---|---|---|---|---|---|---|---|---|---|---|---|---|---|---|
| | | | | Village Ranking | | | | | | | | | | | Villager | Relative | Both |
| Village C, 2013/2014 | 3 | ○ | F | mid | low | low | mid | low | low | high | low | high | high | low | ○ | | |
| | 4 | ○ | M | low | mid | low | high | low | low | mid | mid | high | high | high | ○ | | |
| | 11 | | M | low | high | mid | mid | mid | mid | low | low | low | low | mid | ○ | | |
| | 13 | | M | mid | high | high | mid | high | mid | mid | high | mid | low | mid | ○ | | |
| Total number of rankings | | | | 1 > 21 | 1 > 21 | 1 > 19 | 1 > 20 | 1 > 21 | 1 > 14 | 1 > 13 | 1 > 13 | 1 > 21 | 1 > 12 | 1 > 14 | | | |
| Village Y, 2013/2014 | 18 | ○ | F | low | mid | low | mid | mid | low | low | high | low | low | high | ○ | | |
| | 19 | ○ | M | mid | high | high | high | low | low | low | mid | low | low | high | ○ | | |
| Total number of rankings | | | | 1 > 20 | 1 > 20 | 1 > 19 | 1 > 20 | 1 > 19 | 1 > 15 | 1 > 13 | 1 > 11 | 1 > 20 | 1 > 14 | 1 > 11 | | | |
| Village E, 2014/2015 | 1 | | M | high | high | high | high | low | mid | low | low | mid | low | low | ○ | | |
| | 18 | | F | high | low | low | high | low | low | low | low | low | low | low | ○ | | |
| | 33 | ○ | M | high | mid | high | high | mid | high | low | low | mid | low | mid | ○ | | |
| Total number of rankings | | | | 1 > 57 | 1 > 57 | 1 > 57 | 1 > 55 | 1 > 57 | 1 > 33 | 1 > 13 | 1 > 51 | 1 > 57 | 1 > 12 | 1 > 51 | | | |
| Village E, 2015/2016 | 9 | ○ | M | low | mid | mid | low | high | mid | mid | high | high | high | high | | | ○ |
| | 15 | | M | high | low | mid | mid | mid | low | low | mid | mid | low | high | | ○ | |
| | 16 | ○ | M | mid | high | mid | mid | mid | high | high | mid | high | mid | mid | | ○ | |
| | 24 | | M | high | mid | high | high | mid | high | low | low | low | low | low | ○ | | |
| | 32 | | M | high | mid | high | mid | high | low | low | low | low | low | mid | ○ | | |
| Total number of rankings | | | | 1 > 57 | 1 > 57 | 1 > 57 | 1 > 56 | 1 > 57 | 1 > 33 | 1 > 18 | 1 > 54 | 1 > 57 | 1 > 18 | 1 > 54 | | | |

Source: Created by the author. Note: (1) Household numbers are consistent with those of Figures 2 and 3. (2) The ranking levels in the village are shown in three levels from highest to lowest: high, mid, and low. (3) Households highlighted in grey are mentioned in the text. (4) A table with detailed data is provided in the Appendix A Table A1. (5) ○ indicates as relationship with the person employing *ganyu* (Villager, Relative or Both).

In the above sections, the current research examined households that employed and engaged in *ganyu*. The results showed that these households tended to employ *ganyu* not on the cultivated land owned near the houses where the household lived but on cultivated land located further away (for six out of eight households). However, several other households in the surveyed villages have cultivated land far away from their houses; thus, this was not the only reason for employing *ganyu*. A detailed examination of the households revealed a labour shortage in all six households. In particular, in households with young children, the wife's availability for work might be limited. Moreover, for households with young children, the labour capacity of these young children might also be insufficient, depending on their schooling status. In other words, the six households employed *ganyu*, as it was difficult for them to work in distant fields with the family's limited labour capacity. Employment of *ganyu* to compensate for labour shortages depended on household members but was not limited to female-headed households where male labour was scarce or where the wife's labour was scarce. *Ganyu* was also used to compensate for labour shortages due to husbands' dedication to work other than maize cultivation. Furthermore, hiring relatives for *ganyu* indicated that *ganyu* was used to provide social assistance.

In addition, households with high levels of income and maize self-sufficiency that engaged in *ganyu* indicated that such households engaged in *ganyu* for investment in production. Where households with low levels of income engaged in work that could result in high *ganyu* income, it appears that the livelihood strategy involved supplementing the shortage of cash and maize by engaging in *ganyu*. Furthermore, part of the income to employ *ganyu* was used to address the household's labour shortage. There were also low-income households in which the grandchildren or son, and not the household heads, engaged in *ganyu*. There are other cases in which household members (children and grandchildren) other than the head of the household or spouse were engaged in *ganyu*, especially in the case of households characterized by an elderly head of the household (65 years old or older) (Village E 2014/2015: household number 33, Village E 2015/2016: household number 24). Although the current study assumed that such households both employed and engaged in *ganyu* to compensate for labour and income shortages, it was unclear whether the income from *ganyu* always functioned effectively as a coping strategy against the lack of income.

## 4. Conclusions, Future Challenges, and Prospects

Based on a political economy analysis that incorporates the framework of sustainable rural livelihood, this study aimed to clarify, in detail, the interrelationship between casual wage labour (*ganyu*) and livelihood strategies adopted by small-scale farmers in Malawi. In particular, this study sought to clarify the degrees to which *ganyu* contributed to reducing food insecurity and poverty and tried to reveal the various contexts in which *ganyu* was engaged by rural households.

The study investigated households that engaged in *ganyu*, employed *ganyu*, and both engaged in and employed *ganyu*. The examination of households that engaged in *ganyu* showed that income from *ganyu* was insufficient to supplement the shortage of maize produced in their own households. *Ganyu* does not always function effectively as a coping strategy in this regard. Furthermore, the amount of maize that could be procured differed with the type of *ganyu*. Not all households that wished to work for high-income *ganyu* had the opportunity to engage in such work, and women find it more difficult than men to engage in high-income *ganyu*. However, some women could earn a high income by engaging in *ganyu* for a long period due to their household structures. This suggests that a simple dichotomy based on gender alone is irrelevant in some cases, and detailed information other than the gender of the worker is necessary. These cases also pose an analytical caution that we must investigate significant differences within the gender-based category of 'women' or 'female-headed households'.

Data on the households that employed *ganyu* suggested that some households could procure almost all the maize they required for 12 months if they did not hire *ganyu* and used

the money for their own consumption. These households shared common characteristics, such as the household head was elderly and the amount of their livestock holding assets was relatively large. It appears that they hired *ganyu* to compensate for the labour shortage and that traditional social obligations were involved.

Examination of households that both engaged in and employed *ganyu* showed that the inconvenience of working in distant cultivated lands was the deciding factor behind employing *ganyu*. In addition, households engaged in *ganyu* not only to supplement cash and maize shortages but also to employ labour for their own farm production to compensate for the labour supply shortage in the household. This study observed labour supply shortages among households with female heads, small children, and husbands engaging in off-farm economic activities. Moreover, the fact that relatives were employed as *ganyu* indicated cases of *ganyu* employment for social assistance.

The findings of this study show that previous studies have only partially grasped the actual circumstances of *ganyu* in Malawi. Many farmers engaged in *ganyu* related to agricultural work in neighbouring fields; however, it would be insufficient to consider only such work as *ganyu* and treat *ganyu* as low-wage employment. It is also important to not simply regard households that employ *ganyu* as wealthy but instead explore the impact of *ganyu* on their food consumption and livelihoods in the same way as households that engage in *ganyu*. Furthermore, the actual circumstances of *ganyu* must be acknowledged because each household has its own rationale for engaging in and employing *ganyu*, and these reasons might appear contradictory in the preliminary analysis. In addition, *ganyu* does not always work effectively as a coping strategy to compensate for household income shortages. In the years of crop failure, although the amount of *ganyu* wages did not decrease, the high market price of maize prevented households from purchasing enough food for household consumption.

The political economy analysis adopted in this study revealed important differences and disparities in possession of control over resources (land and labour) and economic activities (maize production and *ganyu* income) between households and within household. Such disparities stemmed from many factors, including gender, age, intra-household relations, spatial mobility of people, geographical and socioeconomic context of village, government policies, and rapidly changing market economy. These factors are interrelated in a complex manner, and together, they shape current and future livelihoods of people in rural Malawi.

With respect to policy implications and recommendations, this study proposes that the overwhelming emphasis in the current policy on increasing agricultural production needs to be reconsidered. Although the current research does not deny the importance of the agricultural sector in reducing poverty and food insecurity, food consumption and nutritional intake in rural households can be enhanced considerably if the policies target not only food production but also off-farm economic activities that would help rural households purchase food. The characteristics and tendencies of the relationship between rural households and *ganyu* shown in this study can act as a reference when determining the types of support and appropriate responses for helping rural households.

A limitation of this study is the same as that of any case study: the study lacks a statistical generalisation of results to the wider population. The current study also lacks a perspective from statistical or econometric analysis due to the small sample size. Therefore, this study does not claim that the result is representative of Malawi in any statistical sense. However, the current research believes that the study results retain utility for analytical generalisation because the research selected case study villages to reflect different socioeconomic characteristics observed in rural Malawi. In any case, the current research call for more studies with household-level perspectives that reflect complex and diverse realities of rural livelihood to be conducted. Such studies would better inform policies related to poverty and food security and provide a balanced view of the dominant macro-perspectives in policymaking not only in Malawi but also in other African countries.

**Author Contributions:** Conceptualisation, H.G.; Methodology, H.G.; Formal analysis, H.G.; Investigation, H.G.; Writing—original draft, H.G and T.T.; Writing—review and editing, H.G., T.T. and D.M. All authors have read and agreed to the published version of the manuscript.

**Funding:** This research was funded by the Japan Society for the Promotion of Science (JSPS) Grants-in-Aid for Scientific Research (KAKENHI) 15K07635 and 20K22151.

**Institutional Review Board Statement:** Not applicable.

**Informed Consent Statement:** Not applicable.

**Data Availability Statement:** Not applicable.

**Acknowledgments:** This work was supported by JSPS Grants-in-Aid for Scientific Research (15K07635 and 20K22151). We are thankful for the cooperation of the local community staff of the Ministry of Agriculture in the Chitipa district and Zomba districts and all members of the study villages. I would also like to express my gratitude to Tetsu Sato (Ehime University) for his project in Malawi (IntNRMS, JPMJSA 1903), which provided us with the opportunity to expand our research concept, and to Atsuko Fukushima (Ehime University), who was a major contributor to the administrative side. This research would not have been possible without the cooperation of these individuals.

**Conflicts of Interest:** The authors declare no conflict of interest.

## Appendix A

**Table A1.** Characteristics of households that both engaged in and employed *ganyu* (Details).

| | (a) | (b) | (c) | (d) (Years) | (e) (People) | (f) (People) | (g) (Ha) | (h) (People/ha) | (i) (KW) | (j) (Days) (Worker) | (k) (Days) (Worker) | (l) (KW/AEU) | (m) (KW/AEU) | (n) (KW/AEU) |
|---|---|---|---|---|---|---|---|---|---|---|---|---|---|---|
| Village C, 2013/2014 | 3 | ○ | F | 57 | 3 | 1.0 | 0.6 | 1.7 | 0 | 288/365 (Head of household) | 1 (Head of household) | 181,463 | 172,000 | 556 |
| | 4 | ○ | M | 35 | 5 | 2.0 | 0.8 | 2.6 | 0 | 327 (Head of household) | 21 (Head of household) | 57,245 | 31,273 | 6114 |
| | 11 | | M | 40 | 7 | 3.0 | 0.6 | 5.2 | 13,150 | 0 | 4 (Head of household) | 15,318 | 0 | 1070 |
| | 13 | | M | 59 | 7 | 5.0 | 0.6 | 8.3 | 17,600 | 240 (Head of household) | 24 (Wife) | 22,311 | 8889 | 2246 |
| Village average | | | | 57 | 6 | 3.2 | 0.6 | 7.5 | 27,164 | | | 44,414 | 24,726 | 2532 |
| Village Y, 2013/2014 | 18 | ○ | F | 39 | 5 | 3.5 | 0.6 | 5.8 | 0 | 48 (Head of household) | 48 (Head of household) | 8638 | 8000 | 6944 |
| | 19 | ○ | M | 49 | 9 | 6.5 | 2.4 | 2.7 | 0 | 0 | 9 (Head of household) | 9063 | 0 | 6301 |
| Village average | | | | 56 | 8 | 4.7 | 1.1 | 5.1 | 95,520 | | | 55,483 | 31,720 | 1635 |
| Village E, 2014/2015 | 1 | | M | 54 | 6 | 2.5 | 1.3 | 1.9 | 11,670 | 0 | 5 (Head of household) | 19,061 | 0 | 194 |
| | 18 | | F | 68 | 3 | 1.5 | 0.7 | 2.1 | 0 | 0 | 2 (Grandchild) | 4240 | 0 | 362 |
| | 33 | ○ | M | 70 | 5 | 2.5 | 0.6 | 4.0 | 47,096 | 0 | 7 (Child) | 9629 | 0 | 1163 |
| Village average | | | | 43 | 5 | 2.2 | 0.5 | 5.5 | 8155 | | | 18,709 | 5774 | 2698 |
| Village E, 2015/2016 | 9 | ○ | M | 32 | 4 | 2.0 | 0.3 | 7.2 | 5375 | 120 (Head of household) | 18 (Head of household) | 68,236 | 47,991 | 9358 |
| | 15 | | M | 53 | 3 | 2.0 | 0.6 | 3.3 | 2822 | 48 (Head of household) | 7 (Head of household)/3 (Wife) | 14,174 | 3505 | 6456 |
| | 16 | ○ | M | 33 | 6 | 2.0 | 0.4 | 4.9 | 10,078 | Year-round (Head of household) | 7 (Head of household)/3 (Wife) | 35,591 | 21,217 | 2334 |
| | 24 | | M | 70 | 5 | 2.5 | 0.7 | 3.7 | 105,720 | 0 | 3 (Child) | 3584 | 0 | 224 |
| | 32 | | M | 56 | 5 | 3.5 | 0.5 | 6.4 | 0 | 0 | 4 (Child) | 8544 | 0 | 2475 |

**Table A1.** *Cont.*

| | (a) | (b) | (c) | (d) | (e) | (f) | (g) | (h) | (i) | (j) | (k) | (l) | (m) | (n) |
|---|---|---|---|---|---|---|---|---|---|---|---|---|---|---|
| | | | | (Years) | (People) | (People) | (Ha) | (People/ha) | (KW) | (Days) (Worker) | (Days) (Worker) | (KW/AEU) | (KW/AEU) | (KW/AEU) |
| Village average | | | | 44 | 5 | 2.3 | 0.7 | 5.0 | 6633 | | | 23,794 | 8599 | 3614 |

Source: Created by the author. Note: (1) Household numbers are consistent with those of Figures 2 and 3. (2) Labour force is denoted as follows: household members aged 16–64 years = 1, household members aged 13–15 years and 65 years or older = 0.5, and household members aged 12 years or younger = 0. (3) Adult equivalent unit (AEU) is denoted as follows: male individual aged 15 years or older = 1, female individual aged 15 years or older = 0.8, and individual aged less than 15 years = 0.5. (4) The livestock holding assets per household, total income per AEU, income from non-agricultural work per AEU, and *ganyu* income per AEU are converted using the rural consumer price index (RCPI), with 2013/2014 values set as 100. In 2013, 1 USD = 369 KW (KW means Malawian Kwacha). (5) Non-agricultural work refers to employed labour (full-time) and self-employed work (does not include *ganyu*). (6) Households highlighted in grey are mentioned in the text. (7) (a) = Household number, (b) = Usage of *ganyu* for far away cultivated land, (c) = Gender of the head of household, (d) = Age of the head of household, (e) = Number of household members, (f) = Labour force, (g) = Area of maize-cultivated land, (h) = Household labour force per area of maize-cultivated land, (i) = Livestock holding assets per household, (j) = Number of days engaged in non-agricultural work, (k) = Number of days engaged in *ganyu*, (l) = Total income per AEU, (m) = Income from non-agricultural work per AEU, (n) = *Ganyu* income per AEU.

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
