# Peer review of "Casual Wage Labour, Food Security, and Sustainable Rural Livelihoods in Malawi"

_sustainability, doi:10.3390/su15075633_

Round 1

Reviewer 1 Report

Overall, I think the paper is well-written and dealing with an interesting and relevant topic. However, I have a few comments for the authors:

1.       The study is lacking literature and Theoretical discussion, the following articles will help the authors to develop these sections.

·         https://doi.org/10.1080/03066150.2019.1708724

·         https://doi.org/10.1080/03768350220150198

·         https://doi.org/10.1177/227797601665873

·         https://doi.org/10.1007/s11135-021-01205-8

·         https://doi.org/10.1080/23311886.2019.1707005

·         https://opendocs.ids.ac.uk/opendocs/handle/20.500.12413/16945

2.       Why did you choose Villages C, Y and E? provide the justification.

3.       The data has been collected from September 2015 in the two northern villages (Villages 144 C and Y) and in August 2016 and August 2017. Did you control for non-response bias, early vs late response bias, and common method bias? Which methods have been applied for this?

4.       Section 2.2 are fragmentary, i.e., “the monetary value of the provided meal 187 was determined” how is the monetary value calculated? Which currency was used? “maize procured by the surveyed households was calculated”, but you calculated this? The method section should discuss in detail how and which methods are used for such calculation and computation. I suggest rewriting this section.

5.       The current findings have not been aligned with existing literature.

6.       The presentations of the tables are confusing, need to revise properly. 

7.       The recommendations and implications of the study are also not written.

Author Response

We thank you for your thoughtful suggestions and insights. The manuscript has benefited from your comments.   The manuscript has been rechecked, and the necessary changes have been made in accordance with your suggestions. Please see the file.   Thank you very much.

Reviewer 2 Report

It is an interesting work focusing on casual wage labour and rural livelihoods from household level. Additional suggestions:

 1. Since the manuscript is using case study approach, it is suggested to apply the case study procedures and refer to some classical case study references, e.g., Yin, 2003 (Case Study Research: Design and Methods).

 2. Addition to case study, it is suggested to combining econometrical analysis to further explorer the mechanism of engaging in Ganyu and employing ganyu in selected area. 

Author Response

(The authors gave the same response as above.)

Reviewer 3 Report

The reviewed article deals with issues that are very important at the local level. It is an interesting case study concerning research at the level of individual households. According to the reviewer, the correct approach to the issues raised requires indicating the importance of casual work on a wider scale (Malawi, Africa, the world). The title of the study does not precisely reflect the research issues. The study needs to refine the methodology (selection of households). Doubts are raised by illegible presentation of data in tables and charts (they are incorrectly constructed). Placing the table on two pages of the study does not make it easier to familiarize with the analyzed issues.

Author Response

(The authors gave the same response as above.)

Reviewer 4 Report

1.  The writing is solid.  The tables not so much.

2.  There is a published agricltural wage. This is not referenced or discussed.  The lowest is for women.  This is not discussed.

Rural MWK1,923.08  Domestic workers MWK1,461.54

https://wageindicator.org/salary/minimum-wage/malawi

3.  Table 1 Villages (columns are OK), but other data should be transposed.  Only show %, but put (N) under each column.

4. Data reduction is not effective.  We need generalizations across Malawi villages, not Village-by-Village data. Unless you are doing case studies, but I think not. 

5. An acceptable revision would have clarified and focused tables.  Too many numbers.  The tables should tell a clear story, elucidating an inspectable pattern of differences.  What is presented here is but one step above a spreadsheet data matrix. 

6. See Hans Zeisel  SAY IT WITH FIGURES

7. Gender issues in wage labor need to be addressed

Author Response

(The authors gave the same response as above.)

Round 2

Reviewer 1 Report

The authors did a good effort to revise the draft, however, the following comments still not addressed.

1.     The authors have substantially used the word “We”, and “Our”. It is advised to avoid such non-academic words and replace them with academic words, such as this research, the current study, etc.

2.     The study is lacking literature and Theoretical discussion. Which underpinning theory (s) have been used to support the current research model? It is suggested that the authors should develop these sections.

3.     The data has been collected from September 2015 in the two northern villages (Villages 144 C and Y) and in August 2016 and August 2017. Did you control for non-response bias, early vs late response bias, and common method bias? Which methods have been applied for this?

4.     The current result and discussion have not been supported and/or aligned with existing literature.

Author Response

Dear. Sir or Madam.

Thank you again for your very kind comments.

Please find attached a table of comment responses, which we hope you will find useful.

Sincerely yours,

Hiroko GONO
